# Markovian Generation Chains in Large Language Models

## Abstract

Large language models (LLMs) have already been widely used in our lives, so what happens when people repeatedly process text using these models? In this paper, we investigate the *Markovian generation chain* in LLMs: a fixed prompt is combined with the most recent output to produce the next output, and this procedure is repeated over multiple iterations. In our simulated iterative generation tasks (e.g., rephrasing and translation), the model's outputs may either converge to a set of similar results or continue to produce distinct outputs for a finite number of steps. While the outcome depends on the model, its configuration, and the input text, it is completely unlike the model collapse observed when models are iteratively trained on generated data. This process can be modeled and analyzed using a Markov chain, and it can be mapped to some real-world scenarios. Our study involves not only various LLMs but also Google Translate as a reference. At the sentence level, LLMs have the potential to increase the text diversity, for example, when the original text shows limited variation.

## 1 Introduction

With the widespread use of large language models (LLMs), the amount of content they generate is also increasing. While much focus has been placed on model collapse caused by the iterative use of generated data in training (Shumailov et al., 2024), our paper concentrates on another perspective: **what happens when content generated by LLMs is re-processed by LLMs?**

Our study can be viewed as a *Markovian generation chain*, since at each step we combine the latest output with the fixed prompt to perform the subsequent inference, and repeat this procedure iteratively. Some researchers have also noted the problems and risks associated with repetitive use of LLMs in inference, such as in translation and rephrasing (Perez et al., 2025; Mohamed et al., 2025). Regarding LLM-generated content, many studies have shown that text from LLMs lacks diversity, but most have focused on word-level, phrase-level, or overall complexity assessments (Chung et al., 2023; Guo et al., 2024). In this paper, we want to analyze diversity at the sentence level by looking at how many distinct sentences are generated, representing a medium-grained granularity.

Our simulation results indicate that it is possible for LLMs to enhance the diversity of text at the sentence level even during iteration generation process, depending on the model, parameters, and the original text. When assessing LLMs, people often pay excessive attention to the outputs, underestimating how much the inputs can shape the results. For example, iteratively generated sentences may enter a state where several similar sentences alternate, or they may produce entirely different content in each round. Therefore, these scenarios are different from, and could even be the opposite of, model collapse. The description of this phenomenon is shown in Figure 1.

Human language has always been changing dynamically (Lieberman et al., 2007; Hamilton et al., 2016). Language evolution and cultural transmission had already been analyzed by researchers before (Griffiths & Kalish, 2007). Given the complexity of the structure of LLMs and the interaction between machines and humans, it is now difficult to strictly interpret what is happening. Instead, we aim to develop a theoretical framework to explain these experimental findings and highlight their potential impact in real-world applications. Combining theory and simulation, we conclude that Markov chains can serve this purpose, which will also help us to better understand the impact of LLMs on language and society.

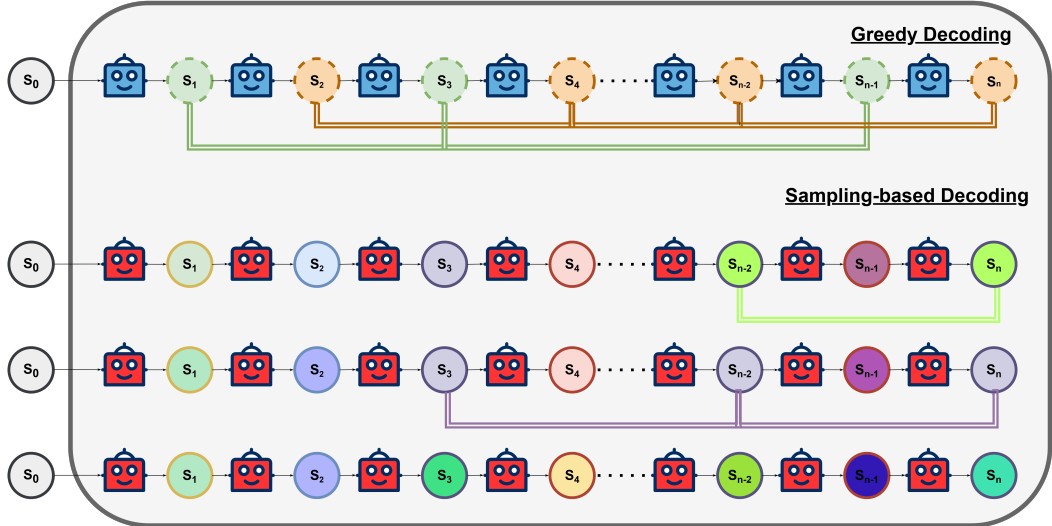

Figure 1: Illustration of our simulation setup for iterative reprocessing using LLMs, i.e., *Markovian generation chains*. Each circle $s_i$ represents the output sentence at iteration $i$ of LLM reprocessing. Sentences sharing the same color and connected to each other indicate exact repetitions observed during iteration. The **Greedy Decoding** regime (top), exhibits high connectivity, with outputs frequently looping back to previously generated sentences, leading to limited diversity. In contrast, under **Sampling-based Decoding** (bottom), while some repetitions and connections remain, the trajectories display more relaxed connectivity and greater sentence-level diversity, reflecting the stochasticity of the decoding process.

## 2 MOTIVATION

LLMs can generate different outputs with the same input in different simulations. This uncertainty is related to various factors, including the hardware setting (He & Lab, 2025), and the most predominant is likely the sampling process (Jang et al., 2016).

For example, the probability $p_i(\tau)$ that token $w_i$ is sampled under the temperature parameter $\tau$ can be expressed as

$$p_\tau(w_i) = \frac{\exp(z_i/\tau)}{\sum_j \exp(z_j/\tau)} \tag{1}$$

where $z_i$ is the logit score for token $w_i$. Therefore, the output is more random at higher temperatures (Holtzman et al., 2019).

Researchers have also developed several sampling methods based on the softmax function above, such as top-k (Fan et al., 2018) and top-p (Holtzman et al., 2019), as well as logit suppression and temperature sampling (Chung et al., 2023), although it may have some bottlenecks (Chang & McCallum, 2022).

The output of LLMs carries some level of uncertainty and randomness, but the model's architecture ensures that the uncertainty and randomness still follow certain statistical patterns. Despite researchers proposing various variants such as multi-token prediction (Gloeckle et al., 2024), the intrinsic properties of the sampling mechanism are preserved.

Hence, the probability of generating the sentence $(w_1, w_2, \ldots, w_n)$ can be expressed as

$$P_\tau(w_1, w_2, \ldots, w_n) = \prod_{t=1}^{n} P_\tau(w_t \mid w_1, \ldots, w_{t-1}). \tag{2}$$

Therefore, we consider that sentence distributions show properties and patterns similar to those of token distributions. We try to approach the problem from a level higher than individual words, as the correspondence between tokens in LLM processing is often not one-to-one.

Sentences are treated as the basic element in this paper, although we also carry out paragraph simulations in the ablation experiments. Further analysis can focus on the transformation probabilities of sentences processed by LLMs, which can be easily mapped to real-world applications such as translation and rephrasing of individual sentences.

Our work centers its analysis and discussion of textual content, with a particular focus on sentences, as language may serve as the first and foundational link in the chain of effects that LLMs have on society. In the experiments, text that has already been processed by LLMs (e.g., through paraphrasing or translation) is iteratively reprocessed, which simulates the real-world phenomenon of text being processed multiple times by LLMs. We believe that the two scenarios we focus on are more common (and maybe also more important) in the real world than model collapse, but research on them is currently lacking. However, given the complexity of LLMs, the properties exhibited by *Markovian generation chain* till need to be verified through simulations.

There has been a lot of discussion about the impact of LLMs on society. For example, LLMs can reshape collective intelligence (Burton et al., 2024) and information may also be distorted during the iterative generation in LLMs (Mohamed et al., 2025). While it may seem straightforward in terms of sentence diversity, very few studies have explored this topic, and none has analyzed the impact of LLM-processed sentences on human language corpora through the lens of Markov chains. Intuitively, it is also easy to establish a **one-to-one** relationship at the sentence level between the original and processed sentences by LLMs.

Some researchers have also used Markov chains to explain the behavior of LLMs, but their approach differs from ours. For example, Zekri et al. (2024) focused on next-token prediction and use Markov chains to analyze repetitive patterns in the output of a single inference process. In comparison, our research centers on sentences, treating the output of one inference as the key element and examining the phenomena of multiple inferences (where each output becomes the input for the next inference).

## 3 SIMULATIONS

Our simulations instantiate an iterative rephrasing pipeline in which an instruction-tuned LLM is repeatedly applied to its own outputs. Each trial begins with an initial text, which is rephrased and then recursively fed back into the model. This loop is unrolled for a fixed number of steps, yielding a trajectory of generations for each seed passage.

**Datasets.** We employ three corpora spanning distinct domains: *BookSum* (narrative prose) (Kryscinski et al., 2022), *ScriptBase-alpha* (cinematic dialogue and stage directions) (Gorinski & Lapata, 2015), and *(BBC) News2024* (contemporary journalism) (Li et al., 2024). From each dataset, we randomly sample 150 documents, and further select the first sentence of each document.

**LLMs.** We evaluate a suite of open instruction-tuned models representative of current architectures: *Mistral-7B-Instruct* (Jiang et al., 2023), *Llama-3.1-8B-Instruct* (Dubey et al., 2024) and *Qwen2.5-7B-Instruct* (Yang et al., 2024). We also conduct simulations with *GPT-4o-mini* (Hurst et al., 2024) through the API.

**Parameters.** Our experiments cover both sampling-based decoding (using the model's default sampling parameters) and greedy decoding, with each trajectory extended for 50 iterations to enable systematic analysis of looping behavior, diversity dynamics, and attractor states. Unless otherwise specified, the temperature used for *GPT-4o-mini* in our sampling-based decoding is 0.7, and the top-p is 0.9.

**Rephrasing Task.** Given a passage, the model is prompted to produce a semantically equivalent paraphrase. The output is then recursively re-input for a total of 50 iterations. This isolates within-language transformation dynamics, allowing us to study whether trajectories collapse to cycles (under greedy decoding) or diversify (under stochastic sampling).

**Translation Task.** Translation can be considered a special case of a rephrasing task. The model is asked to translate the text from English to another language and then back to English. All other settings are kept the same as in the rephrasing task. In addition, we also used the API to run simulations with the non-LLM-powered *Google Translate (v3)* for comparison.

**Prompts.** The prompts used in this paper can be found in the Appendix C.

| Round | Qwen2.5-7B-Instruct | Llama-3.1-8B-Instruct |
|---|---|---|
| 0 | We *begin* with a prologue. | We *begin* with a prologue. |
| 1 | We **start** with a prologue. | The story **commences** with a prologue. |
| 2 | We *begin* with a prologue. | The narrative *begins* with a prologue. |
| 3 | We **start** with a prologue. | The story **commences** with a prologue. |
| 4 | We *begin* with a prologue. | The narrative *begins* with a prologue. |
| 5 | We **start** with a prologue. | The story **commences** with a prologue. |
| 6 | We *begin* with a prologue. | The narrative *begins* with a prologue. |

Table 1: Examples of repeated rephrasing by different LLMs (greedy decoding).

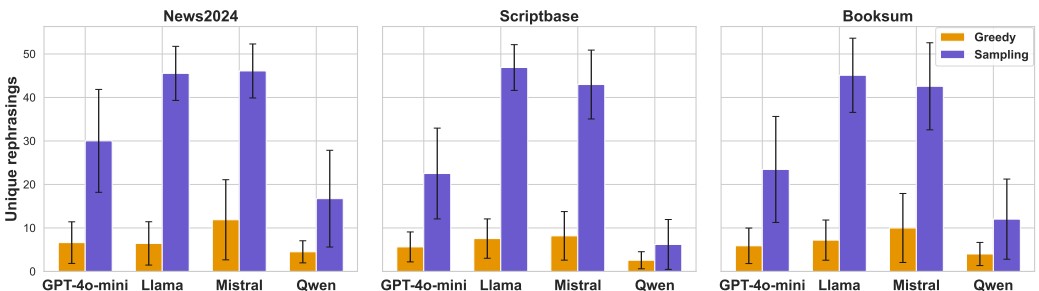

Figure 2: Average number of unique paraphrases generated over 50 iterative rephrasings across three datasets (News2024, Scriptbase, and Booksum), comparing four instruction-tuned LLMs: GPT-4o-mini, Llama-3.1-8B, Mistral-7B, and Qwen-2.5-7B. Results are shown for greedy decoding (orange) and sampling-based decoding (purple). Error bars represent one standard deviation.

# 4 RESULTS ANALYSIS

## 4.1 MAIN RESULTS AND FINDINGS

As previously discussed, LLMs generate outputs with some degree of randomness and uncertainty. Figure 1 provides a schematic diagram of the iterative generation of LLMs in our simulations, including different patterns under different conditions.

In the case of greedy decoding, iteratively generated text is more likely to fall into loops, with a detailed example of greedy decoding shown in Table 1. We can also find that the looped sentences generated by different LLMs differ.

In contrast, with sampling-based decoding, the content generated by LLMs may also fail to converge, at least within the 50 iterations covered in our experiment. For example, Table 2 in the Appendix provides an example in which 50 completely different results were generated over 50 iterations, with the information in the sentences having changed multiple times during the process. LLMs can produce factual drift during text processing, even though we've included the instruction in the prompts. Strictly maintaining consistency between input and output might not truly reflect the real-world dynamics.

The simulation results of the rephrasing task in the three datasets are presented in Figure 2. Despite the differences in the texts of the three datasets, they lead to the same conclusion. We can clearly see that greedy decoding produces more stable outputs, while sampling-based decoding yields higher diversity. Therefore, at the sentence level, the paraphrased content generated by LLMs can either stabilize or diverge after multiple iterations.

We also evaluate text similarity using METEOR (Banerjee & Lavie, 2005), ROUGE-1 (Lin, 2004), and BLEU (Papineni et al., 2002). The results shown in Figure 3, as well as Figures 10 and 11 in the Appendix, suggest that the distribution rapidly reaches a balanced state under greedy decoding. This does not indicate that rephrased sentences are frozen, but that they switch repeatedly among two or more sentences with the similar meaning. The numerical differences between models after stabilization also indicate that the distances between the "similar" sentences generated by different models are not the same.

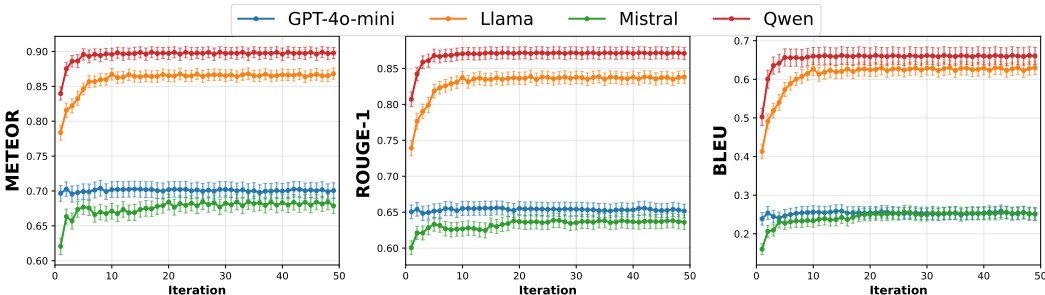

Figure 3: Evolution of text similarity metrics across 50 rephrasing iterations for the BookSum dataset using greedy decoding. Each iteration compares the current rephrased text against the previous iteration's text as reference.

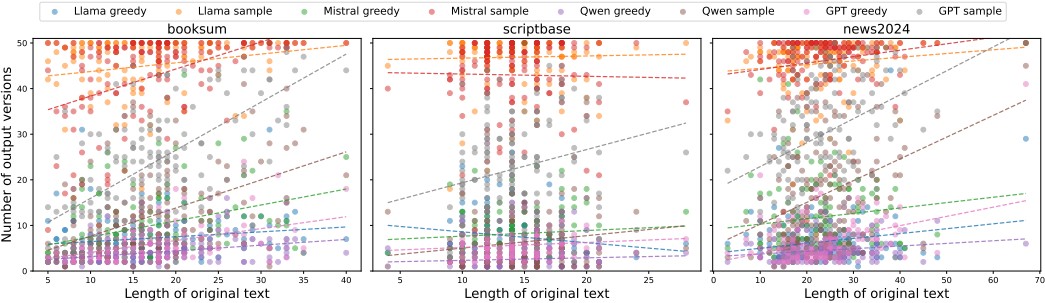

Figure 4: The relationship between the number of sentence versions output by LLMs during 50 iterations and the length of the original text. The x-axis represents the number of words in the original text, and the y-axis represents the number of distinct sentences generated within the first 50 iterations. The dashed line represents the result of linear regression within each group.

## 4.2 PARAMETERS AND INPUTS

Different models and different settings can affect the diversity of LLM output. For instance, the difference between greedy decoding and sampling-based decoding can be considered as a matter of different temperature settings. Figure 2 clearly illustrates the differences across various models, as well as between the two decoding methods.

It is easy to infer from Eq. (2) that increasing the temperature parameter introduces more randomness and produces a wider variety of paraphrased sentences. Figures 12, 13 and 14 in the Appendix further show that as temperature rises, iteratively generated outputs are relatively unlikely to belong to sets of repetitive sentences. Figure 9 in the Appendix presents more results at various temperatures, helping us better understand the effect of temperature on generated patterns.

The behavior of the pattern also depends on the input. Figure 4 indicates a positive correlation between the length of the input sentence and the version number of generated sentences. While this result is intuitive, we should point out that in real-life scenarios, the number of iterative generations is usually limited, and sometimes quite small. As a result, the observed patterns can vary significantly. The detailed numerical results of the correlation analysis are shown in Table 5 in the Appendix.

## 4.3 ABLATIONS: PROMPTS, LONG TEXTS, AND TRANSLATION TASK

We employ various prompts to simulate the rephrasing task with *GPT-4o-mini*, and the results are shown in Figure 5. It's easy to find that the decoding method has a greater influence on the outcome, which may due to the fact that the changes made to the prompt were not substantial enough. Besides, the translation task may also be regarded as a type of rephrasing task when provided with a specific prompt.

To more closely reflect real-world conditions, we perform the simulation where different prompts are alternated, and the results are shown in Figure 6. Although taking different prompts results in

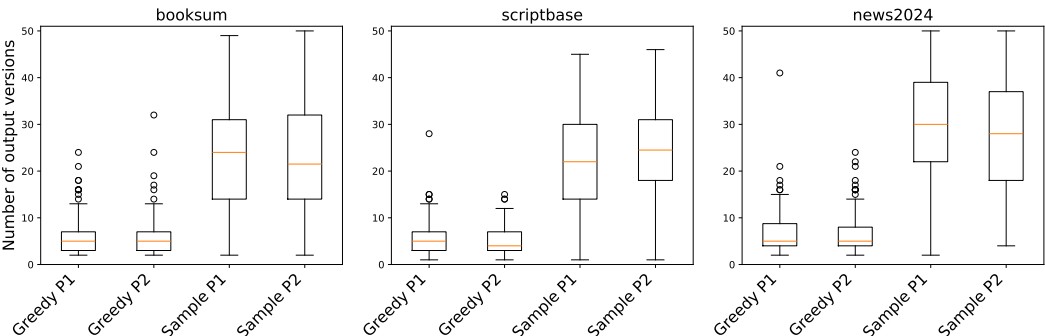

Figure 5: Comparison of the number of sentence versions generated iteratively under different prompts and decoding methods in *GPT-4o-mini* simulations. P1 and P2 correspond to the prompts in Listings 1 and 2 in the Appendix, respectively. The y-axis represents the number of distinct sentences generated within the first 50 iterations. Boxes denote the 25th to 75th percentiles, with the central orange line indicating the median.

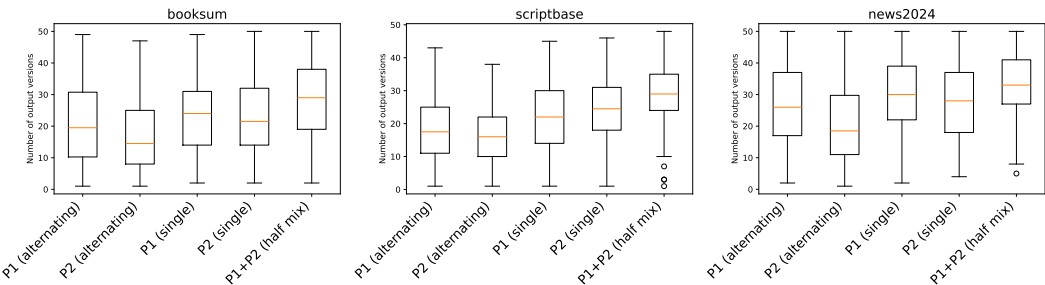

Figure 6: Comparison of results generated by GPT-4o-mini using a single prompt and alternating between two prompts under sampling-based decoding. The last one corresponds to the mixture of the first half of each prompt when alternating between the two prompts.

more diverse outputs, certain sentences will still be repeated. In fact, translation can be seen as a specific case of this scenario.

We also perform simulations using *GPT-4o-mini* (temperature=0.7, top-p=0.9) 450 paragraphs (ranging from 4 to 17 sentences) across three datasets. Certain sentences may repeat across different iterations, as illustrated in Figure 7, but it's uncommon for the whole paragraph to be repeated (i.e., all the sentences in the paragraph are the same). In the 50 iterations, the ratio of the number of sentence versions to the original number of sentences is 26.7 (BookSum), 19.7 (ScriptBase-alpha), 24.2 (News2024).

For the task of translation, we can ask LLMs to repeatedly translate text from English to another language and then back to again. For example, Tables 3 and 4 in the Appendix both present the results of iteratively translating between English and French. While the translation results in Table 3 remain stable, those in Table 4 exhibit a non-periodic cycling behavior.

The translation task has been tested across several language pairs. Since sampling-based decoding is more commonly used than greedy decoding in the real-word scenarios for LLMs, we choose to compare the simulation results using sampling-based decoding with the results from Google Translate. As shown in Figure 8, it is clear that Google Translate produce fewer versions of the sentences. We can also find that translating English into Chinese and then back into English using *GPT-4o-mini* yields more variations of the resulting sentences.

When considering the various scenarios discussed above, it is possible that LLMs could introduce more sentence-level diversity in real-world applications, especially for those accustomed to using tools like Google Translate.

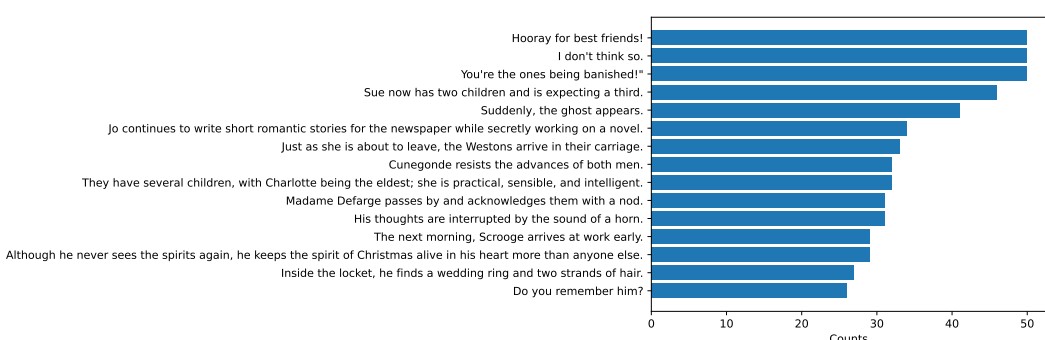

Figure 7: The most frequently repeated sentence after performing 50 simulations (*GPT-4o-mini*, sampling-based decoding, prompt P1) on 150 paragraphs from the Booksum dataset.

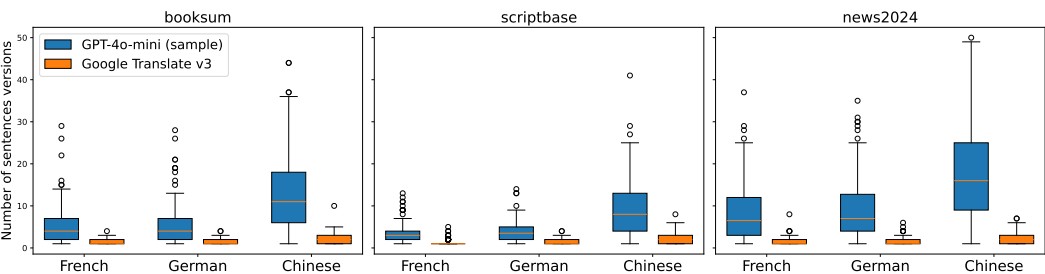

Figure 8: Comparison of the number of sentence versions generated iteratively using *GPT-4o-mini* and *Google Translate (v3)*.

### 4.4 COMPARISON WITH MODEL COLLAPSE

Shumailov et al. (2024) claim that LLMs collapse when trained on recursively generated data, while some researchers argue that the assumptions in model collapse are too strong and can be readily avoided in practice (Schaeffer et al., 2025). Model collapse occurs during the training process, which in most cases is essentially a process of data and information compression. Our paper explores how text and information are transmitted, under the influence of the prior knowledge and patterns inherent to LLMs. Therefore, the scenarios discussed in our paper are different from model collapse, and they may actually be more common in real-world situations.

Previous results imply that even when people repeatedly process text using LLMs, it is possible to maintain a stable distribution of outputs while preserving a certain degree of diversity in the sentence level, which differs from model collapse. The characteristics of the iteratively generated text depend on the model and parameters, as well as the original text. Therefore, we want to propose a theoretical model to explain the existing simulation results, as well as to generalize and make predictions.

## 5 THEORETICAL MODELING

### 5.1 NOTATIONS AND ASSUMPTIONS

If we further consider the limited number of words and the finite sentence length, then we can regard the number of possible sentences as finite. In other words, we can represent the set of sentences in language $l_A$ and language $l_B$ as two discrete state spaces $\mathcal{S}_A$ and $\mathcal{S}_B$:

$$\mathcal{S}_A = \{a_1, a_2, \ldots, a_m\} \tag{3}$$
$$\mathcal{S}_B = \{b_1, b_2, \ldots, b_n\}. \tag{4}$$

For the rephrasing task, we provide a given sentence and design a prompt, then have the LLMs rewrite it. This process can be represented as follows:

$$a \xrightarrow{\text{prompt}} a', \qquad a, a' \in \mathcal{S}_A \tag{5}$$

where $p$ denotes all parameters involved in the process, including model configurations and prompt design.

For the translation task, we have the similar expression:

$$a \xrightarrow{\text{prompt}} b, \qquad a \in \mathcal{S}_A, \quad b \in \mathcal{S}_B . \tag{6}$$

In our experiments, the outputs of LLMs are only dependent on the last state, which naturally leads us to consider using Markov chains to model this process. Below, we use the task of machine translation as an example. The task of rephrasing can be regarded as a degenerate case of machine translation. Such a setup also applies perfectly to paragraphs or articles.

Let $M_p$ denote the model $M$ used for translation from language $l_A$ to language $l_B$ with parameter $p$. Therefore, the transition matrix of the translation procedure could be expressed as:

$$[\mathbf{P}_{AB}(M_p)]_{ij} = \Pr(b_j | a_i, M_p) . \tag{7}$$

When the sentence is translated back from language $l_B$ to language $l_A$ using the same model $M$ with parameter $p$, the transition matrix $\mathbf{P}_{ABA}(M_p)$ can be formulated as:

$$\mathbf{P}_{ABA}(M_p) = \mathbf{P}_{AB}(M_p)\mathbf{P}_{BA}(M_p) . \tag{8}$$

Therefore, the multi-turn translation process can be described by

$$\mathbf{s}_n = \mathbf{s}_0 [\mathbf{P}_{ABA}(M_p)]^n \tag{9}$$

where $\mathbf{s}_0$ represents the initial state and $\mathbf{s}_n$ represents the results after $n$ iterations.

We can also more generally use $\mathbf{P}(M_p)$ to denote the transition matrix. In translation tasks, it refers to the process of translating text into another language and then back into the original language, while in paraphrasing tasks, it involves only a single act of rephrasing.

## 5.2 Interpretation of Simulation Results

Next, we define the sentences within the loop as $C_i$. For sentences in $C_i$, given the model, parameters, and prompts, their processed results also remain within $C_i$. We also use $T$ to represent all sentences in transient states, meaning that these sentences will eventually converge to some $C_i$ after a number of translations. To better represent different scenarios, we can write the transition matrix in a more specific way:

$$\mathbf{P}(M_p) = \begin{bmatrix} \mathbf{P}_1 & \mathbf{0} & \cdots & \mathbf{0} & \mathbf{0} \\ \mathbf{0} & \mathbf{P}_2 & \cdots & \mathbf{0} & \mathbf{0} \\ \vdots & \vdots & \ddots & \vdots & \vdots \\ \mathbf{0} & \mathbf{0} & \cdots & \mathbf{P}_c & \mathbf{0} \\ \mathbf{Q}_1 & \mathbf{Q}_2 & \cdots & \mathbf{Q}_c & \mathbf{P}_T \end{bmatrix} \tag{10}$$

where $\mathbf{P}_i$ stands for the transition matrix inside $C_i$, $\mathbf{Q}_i$ is the transition matrix from $T$ to $C_i$, and $\mathbf{P}_T$ represents the transition matrix for sentences within $T$.

This model can explain many previously observed experimental results. For example, starting from the same initial state, simulations could ultimately lead to different stable distributions. In our theoretical model, a sentence in set $T$ may be attracted to more than one $C_i$.

As we have presented before, the translation outcomes become stable after several iterations, because they are attracted to some $C_i$. Since a $C_i$ may contain multiple sentences, we can also observe the loop of similar sentences, as listed in Figure 9, Tables 3 and 4.

As we can find in the definition, $\mathbf{P}(M_p)$ is determined by the model and its parameters, for example, the *temperature*. Higher *temperature* increases the likelihood of transitioning to a farther $C_i$, resulting in worse outcomes after converge.

When $P_T$ makes up most of the matrix in Eq. (10), implying that the cycles contain only a few sentences, which corresponds to a divergent scenario where new sentences emerge in each of 50 iterations.

### 5.3 ENTROPY

Assuming sentence $S$ is a random sentence from the original data or in the output of LLMs, then its corresponding entropy can be expressed as:

$$\mathrm{H}(S) := -\sum_{s \in \mathcal{S}} p(s) \log p(s) \tag{11}$$

where $p(s)$ is the probability of sentence $s$ in the set of all possible sentences $\mathcal{S}$.

For convenience, we denote the probability distribution of sentence $S$ as $X$ and $P$ as the probability transition matrix of one LLM processing step. Then we use $\mathrm{H}(X)$ and $\mathrm{H}(XP)$ to represent the corresponding entropy.

As the previous entropy function is Schur-concave function, then if P is doubly stochastic matrix, we have

$$\mathrm{H}(XP) \geq \mathrm{H}(X). \tag{12}$$

In LLMs, the transition matrix P is not necessarily doubly stochastic. Therefore, the above inequality does not always hold in practice.

This also illustrates that, theoretically, the entropy of sentences processed by LLMs does not always decrease. The entropy may increase or decrease depending on $P$ and $X$. In fact, owing to the diversity of LLM outputs, their entropy can actually increase. For instance, the same original sentence may be transformed into multiple possible alternatives.

### 5.4 CROSS-ENTROPY

Assuming $X$ and $Y$ are two different distributions, and $P$ still represents the state transition matrix of the Markov chain, we arrive at the following inequality concerning the Kullback-Leibler (KL) divergence:

$$D_{\mathrm{KL}}(XP \parallel YP) \leq D_{\mathrm{KL}}(X \parallel Y). \tag{13}$$

The detailed proof, which depends on the properties of P, is provided in the Appendix B.

Therefore, for a distribution $\pi$ that satisfies $\pi P = \pi$, we have the following expression for $n$ iterations,

$$D_{\mathrm{KL}}(XP^n \parallel \pi) \leq D_{\mathrm{KL}}(XP \parallel \pi) \leq D_{\mathrm{KL}}(X \parallel \pi). \tag{14}$$

Therefore,

$$\mathrm{H}(XP^n) - \mathrm{H}(X) \geq \mathbb{E}_X[\log \pi] - \mathbb{E}_{XP^n}[\log \pi]. \tag{15}$$

In other words, $XP^n$ is closer to $\pi$ in terms of distribution compared to $X$. After repeated iterations, the distribution of LLM-processed sentences converges to that stationary distribution $\pi$.

### 5.5 MIXTURE

In the real world, it is also common for pre-processed and post-processed texts to be mixed together. Therefore, we also analyzed the diversity in this scenario.

For instance, we can define the distribution $Y$ of the mixed text as:

$$Y = bX + (1-b)XP \tag{16}$$

where $b$ is the mixed coefficient and satisfies $0 < b < 1$.

Therefore, the entropy of the mixture between the two can be bounded by the following inequality:

$$bH(X) + (1-b)H(XP) \leq H(Y) \leq bH(X) + (1-b)H(XP) + h(b) \tag{17}$$

where

$$h(b) = -b \log b - (1-b) \log(1-b). \tag{18}$$

It is easy to see that $0 < h(b) \leq \log 2$ and $h(b)$ reaches its maximum when $b = 0.5$.

Through repeated iterations and mixing, the entropy remains bounded. Therefore, our earlier conclusions still hold true in mixed scenarios.

### 5.6 Complex Scenarios in Reality

The reality is often more complex, for example, people use different prompts and different LLMs. What we're trying to model is the situation where different users apply the same prompts to the same text repeatedly, such as in translation tasks, which is something that often happens in real-world applications.

Different prompts can readily lead to different outcomes. It's easy to imagine that different LLMs converge in different cycles, and a clear example can be seen in Table 1. Thus, in real-world applications, texts processed by LLMs tend to exhibit greater diversity in the sentence level.

Moreover, human-in-the-loop situations should be considered (Chung et al., 2023), such as when people edit or modify content generated by LLMs (Geng & Trotta, 2025). This kind of change will further affect the distribution of LLM-processed text in the real world.

## 6 Related Work

**Model Collapse.**   As noted above, model collapse occurs during the process of iteratively training new models with synthetic data (Shumailov et al., 2024; Guo et al., 2023). The occurrence relies on strong assumptions, and there are various mitigation solutions (Gerstgrasser et al., 2024). For instance, a mixture of real and synthetic data is likely to avoid model collapse (Seddik et al., 2024). Similar collapse risks may also spread further into the field of knowledge (Peterson, 2025).

**LLM-Generated Content.**   Many researchers have noted the low diversity of content generated by LLMs. For example, Padmakumar & He (2023) have pointed out that writing with LLMs can reduce content diversity, and Xu et al. (2025) measured the issue of insufficient plot diversity. Diversity can be evaluated beyond text, such as in code (Shypula et al., 2025).

**Language Convergence.**   Kandra et al. (2025) find that LLM will adapt their language use according to their conversational partner, i.e., another LLM. Human users of LLMs have likely learned language from the models (Yakura et al., 2024; Geng et al., 2024), and LLMs tend to show linguistic convergence in their communication with humans (Blevins et al., 2025).

## 7 Discussions and Conclusions

Culture mediated or generated by machines has already caught the interest of researchers (Brinkmann et al., 2023). Due to the widespread adoption of LLMs and their characteristics, which differ from earlier tools (e.g., Google Translate), the patterns of information transmission and knowledge iteration may have shifted.

With the growing volume of text processed by LLMs, we want to address the issue of iterative generation. Hence, we apply the *Markovian generation chains* in the simulations to model realistic scenarios. For instance, in reality, a text is handled with LLMs by one person and subsequently processed again with LLMs by someone else.

Although LLMs may reduce diversity and quality of used with large amounts of high-quality text, they could conversely increase it when applying to low-quality texts, even after repeated iterations. In our simulations, LLMs may generate new sentences through a wider variety of combinations. When evaluating the impact of LLMs on text processing, the quality of the input texts cannot be ignored.

Given the nature of LLMs, we chose to use Markov chains to interpret the simulation results. We speculate that, for a given sentence, LLMs consider the meaning of several other sentences to be similar to it. Consequently, any of these sentences could appear in the output with varying probabilities. Therefore, LLMs have the potential to enrich original expressions at the sentence level.

Our research offers a new perspective on the features of LLMs, but it does not suggest that this is a drawback of LLMs. In addition, we think that researchers can assess the stability and creativity of LLM outputs through sentence-level diversity, as it is at least an intuitive and transparent metric.

## ETHICS STATEMENT

This paper does not involve human participants or sensitive data sets, so it should not raise any ethical concerns.

## REPRODUCIBILITY STATEMENT

The main parameters have already been mentioned in the main text, while the appendix provides supplementary theoretical proofs and additional results. The data we used have also been uploaded. If the paper is accepted, we will also make the code public.

Despite being closed-source models, we incorporate *GPT-4o-mini* and *Google Translate* via their APIs, due to their widespread use.

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

## A THE USE OF LLMS

LLMs were used to help improve text and optimize code. Moreover, we took advantage of their search functionality as a complement to Google Search and also attempted to have LLMs assess potential issues in some of our arguments. We recognize the shortcomings of LLMs, so we treat them solely as an aid.

## B CONTRACTION OF RELATIVE ENTROPY

Kullback-Leibler (KL) divergence is defined as,

$$D_{KL}(X\|Y) = \sum_i X_i \log\left(\frac{X_i}{Y_i}\right). \tag{19}$$

Similarly,

$$D_{KL}(XP\|YP) = \sum_j (XP)_j \log\left(\frac{(XP)_j}{(YP)_j}\right) \tag{20}$$

where

$$(XP)_j = \sum_i x_i P_{ij}. \tag{21}$$

Then,

$$D_{KL}(XP\|YP) = \sum_j \left(\sum_i X_i P_{ij}\right) \log\left(\frac{\sum_i X_i P_{ij}}{\sum_i Y_i P_{ij}}\right). \tag{22}$$

Then according to Jensen's inequality,

$$D_{KL}(XP\|YP) \le \sum_j \left(\sum_i X_i P_{ij} \log \frac{X_i P_{ij}}{Y_i P_{ij}}\right) = \sum_i X_i \log \frac{X_i}{Y_i} \sum_j P_{ij}. \tag{23}$$

Given that $P$ is a stochastic matrix, we have $\sum_j P_{ij} = 1$. Therefore,

$$D_{KL}(XP \parallel YP) \le D_{KL}(X \parallel Y). \tag{24}$$

## C PROMPTS

Listing 1: Prompt for rephrasing

```
"Given a passage, rephrase it while preserving all the original
    meaning and without losing any context.\n"
"Do not write an introduction or a summary. Return only the rephrased
    passage.\n\n"
"Rephrase the following text:\n{content}"
```

Listing 2: Prompt for rephrasing (ablation)

```
"Rephrase the following text:\n{content}"
```

Listing 3: Prompt for translation

```
"Translate the following English text into {target_lang}:"
"Translate the following {target_lang} text into English:"
```

## D OTHER RESULTS

| Round | Llama-3.1-8B-Instruct | Qwen2.5-7B-Instruct |
|---|---|---|
| 0 | We begin with a prologue. | We begin with a prologue. |
| 1 | The narrative commences with a prologue. | We start with a prologue. |
| 2 | The story begins with a prologue that sets the scene. | We begin with a prologue. |
| 3 | The narrative commences with a prologue that establishes the backdrop for the story. | We start with a prologue. |
| 5 | The story starts with a prologue, which establishes the setting and lays the groundwork for the events that unfold in the narrative. | We start with a prologue. |
| 10 | The narrative begins with a prologue, which sets the scene and lays the groundwork for the events that unfold in the story that follows. | We begin with a prologue. |
| 20 | The narrative begins with a prologue, establishing the groundwork and setting the scene for the events that will unfold throughout the story. | We begin with a prologue. |
| 30 | The narrative begins with a prologue that sets the scene and lays the groundwork, providing a solid foundation for the events that follow to unfold and develop. | We begin with a prologue. |
| 40 | The narrative begins with a prologue that sets the scene, provides essential background information, and lays the groundwork, establishing a solid foundation that allows the subsequent events to unfold in a logical and cohesive manner. | We begin with a prologue. |
| 49 | The story starts with a prologue that lays the groundwork by providing key background information and establishing the setting, thus creating a solid foundation that allows the subsequent events to unfold in a logical and coherent manner. | We start with a prologue. |
| 50 | The story begins with a prologue that sets the stage by offering essential background information and establishing the setting, thereby laying a solid foundation that enables subsequent events to unfold in a logical and coherent way. | We begin with a prologue. |

Table 2: Examples of repeated rephrasing by different LLMs (temperature=0.7, top-p=0.9).

| Round | French | English |
|---|---|---|
| 0 | | We begin with a prologue. |
| 1 | Nous commençons par un prologue. | We begin with a prologue. |
| 2 | Nous commençons par un prologue. | We begin with a prologue. |
| 3 | Nous commençons par un prologue. | We begin with a prologue. |
| 4 | Nous commençons par un prologue. | We begin with a prologue. |

Table 3: Examples of iterative translation via GPT-4o-mini (temperature=0.7, top-p=0.9).

| Round | French | English |
|-------|--------|---------|
| 0 | | Elizabeth **reads** through Darcy's letter with a mixture of emotions. |
| 1 | Elizabeth **lit** la lettre de Darcy avec un mélange d'émotions. | Elizabeth read Darcy's letter with a mix of emotions. |
| 2 | *Elizabeth a lu la lettre de Darcy avec un mélange d'émotions.* | *Elizabeth read Darcy's letter with a mix of emotions.* |
| ⋮ | ⋮ | ⋮ |
| 11 | Elizabeth **lut** la lettre de Darcy avec un mélange d'émotions. | Elizabeth read Darcy's letter with a **mixture** of emotions. |
| 12 | Elizabeth a lu la lettre de Darcy avec un mélange d'émotions. | Elizabeth read Darcy's letter with a **mixture** of emotions. |
| ⋮ | ⋮ | ⋮ |
| 19 | Elizabeth **lut** la lettre de Darcy avec un mélange d'émotions. | Elizabeth read Darcy's letter with a mix of emotions. |

Table 4: Examples of iterative translation via GPT-4o-mini (temperature=0.7, top-p=0.9). The omitted lines are all identical to the content in the second round (the italicized sentences). The similar scenario also occurs after the 19th round.

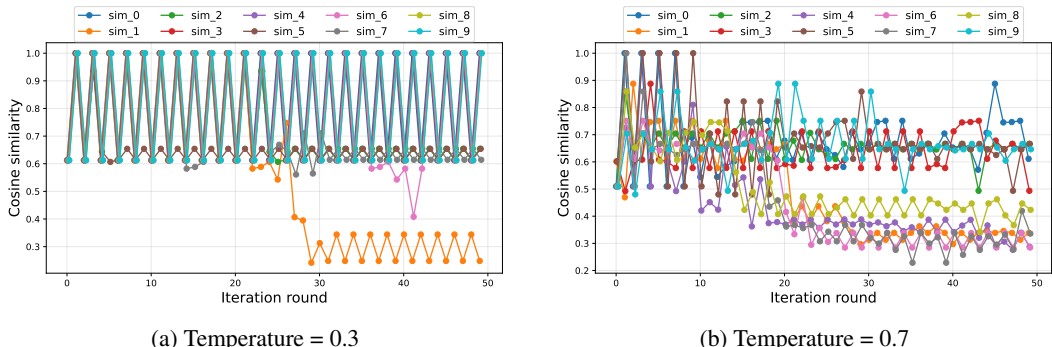

(a) Temperature = 0.3                    (b) Temperature = 0.7

Figure 9: Simulation results of GPT-4o-mini for the same sentence with different parameters. Each line represents the result of a simulation consisting of 50 consecutive iterations. The x-axis indicates the number of rephrases, and the y-axis indicates the cosine similarity with the original text, which compares the angle between TF-IDF (term frequency–inverse document frequency) vectors (2-gram to 4-gram).

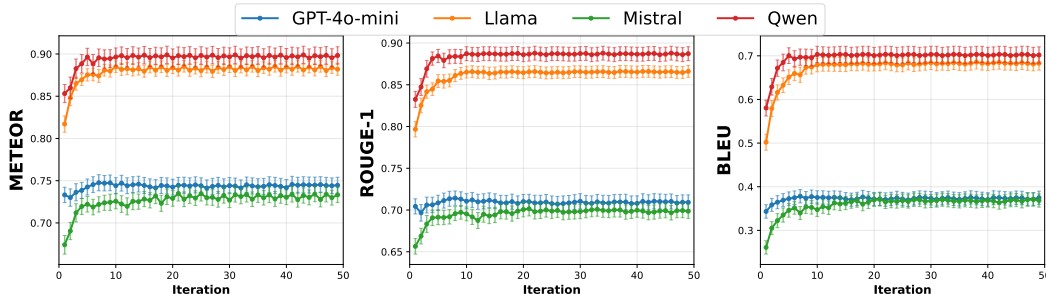

Figure 10: Evolution of text similarity metrics across 50 rephrasing iterations for the News2024 dataset using greedy decoding. Each iteration compares the current rephrased text against the previous iteration's text as reference.

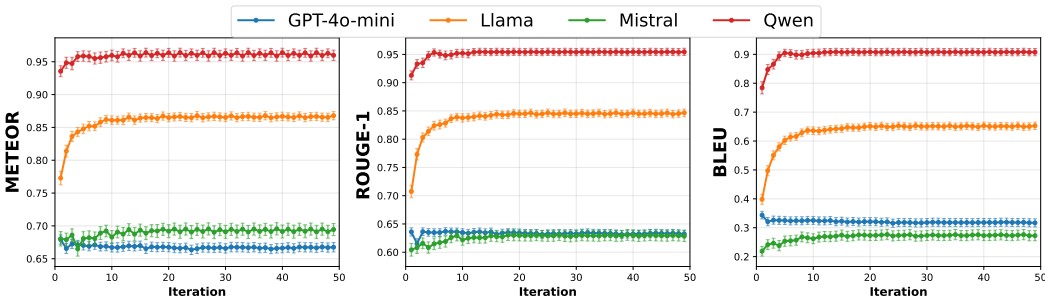

Figure 11: Evolution of text similarity metrics across 50 rephrasing iterations for the ScriptBase dataset using greedy decoding. Each iteration compares the current rephrased text against the previous iteration's text as reference.

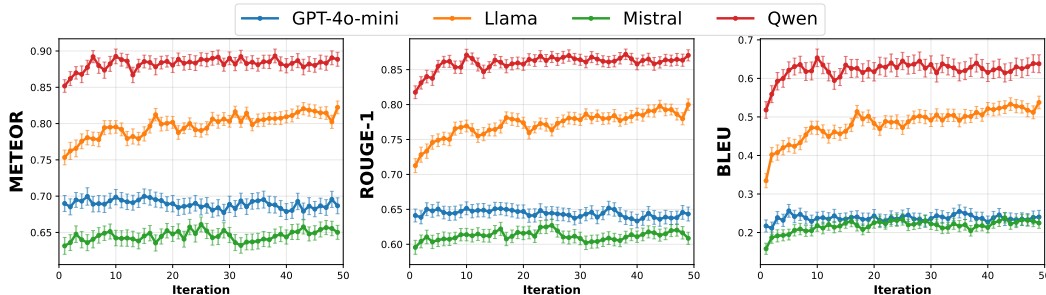

Figure 12: Evolution of text similarity metrics across 50 rephrasing iterations for the BookSum dataset using sampling-based decoding. Each iteration compares the current rephrased text against the previous iteration's text as reference.

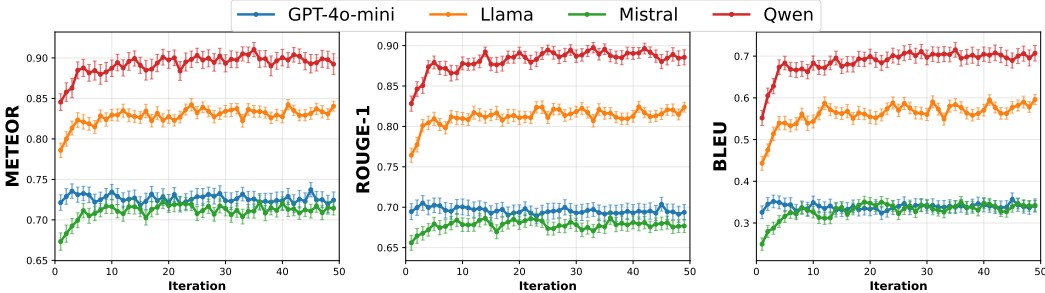

Figure 13: Evolution of text similarity metrics across 50 rephrasing iterations for the News2024 dataset using sampling-based decoding. Each iteration compares the current rephrased text against the previous iteration's text as reference.

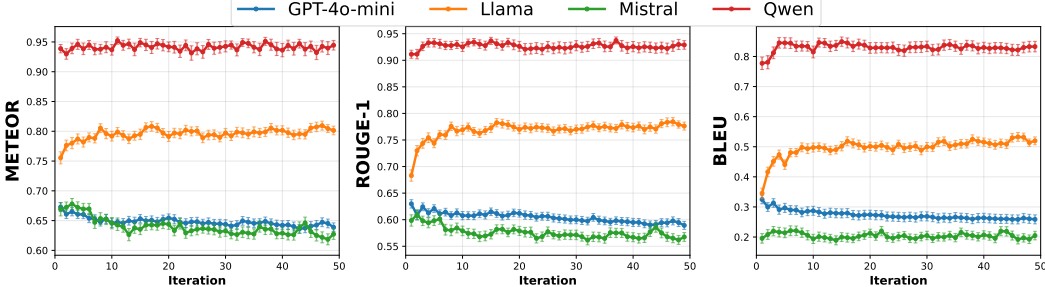

Figure 14: Evolution of text similarity metrics across 50 rephrasing iterations for the ScriptBase dataset using sampling-based decoding. Each iteration compares the current rephrased text against the previous iteration's text as reference.

| Dataset | Model | r | p | $R^2$ | Slope |
|---|---|---|---|---|---|
| booksum | Llama greedy | 0.171 | 3.611e-02 | 0.029 | 0.107 |
| | Llama sample | 0.166 | 4.272e-02 | 0.027 | 0.191 |
| | Mistral greedy | 0.324 | 5.135e-05 | 0.105 | 0.350 |
| | Mistral sample | 0.438 | 2.046e-08 | 0.192 | 0.594 |
| | Qwen greedy | 0.353 | 9.427e-06 | 0.125 | 0.125 |
| | Qwen sample | 0.494 | 1.330e-10 | 0.244 | 0.616 |
| | GPT greedy | 0.476 | 7.658e-10 | 0.226 | 0.261 |
| | GPT sample | 0.638 | 1.542e-18 | 0.407 | 1.054 |
| scriptbase | Llama greedy | -0.174 | 3.338e-02 | 0.030 | -0.238 |
| | Llama sample | 0.030 | 7.150e-01 | 0.001 | 0.048 |
| | Mistral greedy | 0.073 | 3.731e-01 | 0.005 | 0.124 |
| | Mistral sample | -0.021 | 8.012e-01 | 0.000 | -0.050 |
| | Qwen greedy | 0.094 | 2.510e-01 | 0.009 | 0.056 |
| | Qwen sample | 0.157 | 5.479e-02 | 0.025 | 0.272 |
| | GPT greedy | 0.105 | 2.017e-01 | 0.011 | 0.109 |
| | GPT sample | 0.231 | 4.471e-03 | 0.053 | 0.727 |
| news2024 | Llama greedy | 0.187 | 2.216e-02 | 0.035 | 0.107 |
| | Llama sample | 0.114 | 1.659e-01 | 0.013 | 0.081 |
| | Mistral greedy | 0.111 | 1.757e-01 | 0.012 | 0.118 |
| | Mistral sample | 0.197 | 1.585e-02 | 0.039 | 0.141 |
| | Qwen greedy | 0.199 | 1.454e-02 | 0.040 | 0.058 |
| | Qwen sample | 0.373 | 2.600e-06 | 0.139 | 0.477 |
| | GPT greedy | 0.367 | 3.765e-06 | 0.135 | 0.203 |
| | GPT sample | 0.385 | 1.116e-06 | 0.149 | 0.525 |

Table 5: Results of the Correlation Analysis. $r$ represents the Pearson correlation coefficient, and $R^2$ represents the coefficient of determination.

