# OpenReview forum: "Markovian Generation Chains in Large Language Models"
_ICLR.cc/2026/Conference — Submitted to ICLR 2026_

### Official Review · Reviewer_Rz5T · 2025-10-27

**Soundness:** 3
**Presentation:** 3
**Contribution:** 1
**Rating:** 2
**Confidence:** 2

**Summary:**

This paper investigates repeated inference on textual data using LLMs, where the output of one turn is fed back as input for the next inference step. This is tested empirically via a rephrasing and translation task, and theoretically investigated via a proposed markov chain framework.

The authors find that the iterative process behaves very differently depending on the decoding strategy, and conclude that this iterative inference process is unlike model collapse due to iterative training on repeated generated LLM data. Indeed, the paper finds that, contrary to some concerns, LLMs can potentially increase text diversity at the sentence level through this iterative sentence processing.

**Strengths:**

The paper is easy to follow and results/math is presented in a pedagogical way.
Interesting dichotomy between iterative inference with LLMs, and the more common modal collapse where models are recursively trained on LLM generated data.

**Weaknesses:**

**Concern 1:**

My main concern is regarding the significance of the contribution.
Especially in relation to the scope and complexity of the experiments (See concern 2).

The results and findings from the experiments seem very intuitive, and almost self-evident. For example, since greedy decoding is deterministic and the number of possibilities relevant outputs are finite, we are bound to end up in a loop? Similarly, the more randomness/temperature the more different outputs one would expect.

This also goes for the discovered relationship between text length and the number of unique phrasings/translations. This also risks being so intuitive, that it might come across as non-interesting.

I will note that something does not need to be surprising to be informative. It could therefore be that these findings are crucial to the research community although not to me personally (hence my low confidence score in the review), but I was not convinced by the motivation in the paper. (See concern 3)


**Concern 2:**

The scope and complexity of the experiments seem (to me) to be too limited, to be of much interest. The paper focuses on single sentences in the current day where LLMs are capable of vast context sizes and generating long stories, programming projects, etc… So whilst it might be that recursive inference on LLM generated output leads to something interesting, I think readers would be more convinced if the scope of the experiments was increased.

I would suggest strengthening the appeal of the paper by perhaps setting up chains of agentic LLMs that pass information from one point to another. One could track how that initial information transforms, as more and more LLMs interpret/rephrase the information as they propagate it forward. Or something similar, where the scope is not only iterative transformations to individual sentences.

**Concern 3:**

The current motivation for why this task is important could use some strengthening. Some statements come across a bit hand-wavey, rather than feeling properly empirically/theoretically grounded. For example:

Line 23: “Researchers need to think more about where this chain will lead.”
This sentence by itself is no motivation for why researchers should think about where this chain leads.

Line 109: “... which can be easily mapped to real-world applications such as translation…”
How exactly is this easily mapped to real-world scenarios? What scenario do you have in mind where LLM agents will, on the sentence level, transform/translate text during inference?

Line 113: “... which simulates the real-world phenomenon of text being processed multiple times by LLMs”
Similar to line 109 (and concern 2) I fear the reader might not trivially be able to see how your experiments map to real-world scenarios.

My suggestion would be to incorporate some of the final paragraphs in discussion and conclusion (Line 482) to an earlier part of the paper.

**Questions:**

Do you imagine it possible to expand your study/frameork from single-sentence transformations, to more complex and larger units of data?

---

> ### Author Response · Authors · 2025-11-21
>
> Thank you for your very detailed feedback and questions. We hope the following answers will address your concerns, and we will also provide more clarifications and additional information in future versions.
>
> **Weakness 1**: My main concern is regarding the significance of the contribution. Especially in relation to the scope and complexity of the experiments …
> **Answer**: Thank you for your feedback. We agree that the majority of the simulation results were within our expectations, but **not in every case**. For example, in our experiments, LLMs may produce different outputs in the first fifty iterations without entering a cycle (Table 2). There may be more than two stable states within the cycle (Table 4). These phenomena may not align with everyone’s predictions. As the structure of LLMs is quite complex, even **simple ideas require simulation to be validated**.
>
> **Weakness 2**: The scope and complexity of the experiments seem (to me) to be too limited, to be of much interest.
> **Answer**: We would like to highlight that most prior research has concentrated on either the macro level (such as paragraphs and entire documents) or the micro level (such as words and short phrases), with very few studies addressing diversity at the sentence level. At the sentence level, we can establish Markov chains for explanation and illustration. But at the level of words or phrases, it is difficult to establish a one-to-one correspondence. For long texts or paragraphs, it is challenging to observe such correspondence within a limited number of experiments.
> We are very grateful for your suggestion about the chains of agentic LLMs, as it provides a much broader context. Our paper can be considered a brief exploration within that larger framework.
>
> We have also supplemented the simulation with **alternating rephrasings using different prompts**, and the results are similar, where cycles appear but the states (sentences) differ within those cycles. We will update these results in the next version.
>
> **Weakness 3**: The current motivation for why this task is important could use some strengthening. Some statements come across a bit hand-wavey, rather than feeling properly empirically/theoretically grounded.
> **Answer**: We appreciate your question and will revise certain expressions based on your feedback.
>
> Line 23: The term “chain” in the sentence has a double meaning: it refers both to the Markovian generative chain mentioned in the title, and to the chain reaction that results from the real-world application of LLMs.
> Line 109 & 113: Thank you for highlighting this. We will make the appropriate adjustments. We agree with your point that paraphrasing and translation between LLM agents should involve longer texts. Our experiments simulate a real-life scenario where different people use LLMs to paraphrase and translate a sentence.
> Line 482: Thank you for your suggestion. We will revise this part.
>
> **Question**: Do you imagine it possible to expand your study/framework from single-sentence transformations, to more complex and larger units of data?
> **Answer**: Yes, in the past few days, we have done **new simulations on 450 paragraphs (ranging from 4 to 17 sentences)** across three datasets. For example, in simulations using GPT-4o-mini on paragraphs (temperature=0.7, top-p=0.9), certain sentences may repeat across different iterations. But it's uncommon for the whole paragraph to be repeated (i.e., all the sentences in the paragraph are the same). In the 50 iterations, the ratio of the number of sentence versions to the original number of sentences is 26.7 (BookSum), 19.7 (ScriptBase-alpha), 24.2 (News2024).
>
> We welcome any further questions or comments you may have\!

---

### Official Review · Reviewer_svf8 · 2025-10-27

**Soundness:** 2
**Presentation:** 3
**Contribution:** 1
**Rating:** 2
**Confidence:** 4

**Summary:**

This paper investigates the Markovian generation chain phenomenon in large language models (LLMs), where a model’s output is recursively fed back as the next input together with a fixed prompt. Through iterative rephrasing and translation tasks, the authors observe that outputs may either converge to a limited set of sentences (under greedy decoding) or remain diverse for multiple steps (under sampling). They formalize this process as a finite-state Markov chain and provide a theoretical analysis using entropy and KL-divergence properties to explain convergence patterns.

The paper claims to offer a new perspective for understanding repeated inference of LLMs and potential implications for text diversity and societal effects.

**Strengths:**

1. Clear presentation and solid structure — the manuscript is well written and easy to follow.

2. Systematic empirical setup — experiments cover multiple datasets (BookSum, ScriptBase, News2024) and several open models (Llama-3.1-8B, Mistral-7B, Qwen-2.5-7B, GPT-4o-mini).

3. Sound use of Markov-chain formalism — modeling iterative generation as a Markov process is mathematically reasonable and consistent with existing literature (e.g., Zekri et al., 2024).

**Weaknesses:**

1. Lack of novelty.
The main idea—treating LLM iterative inference as a Markov process—has already appeared in prior studies such as Zekri et al. (2024) “Large Language Models as Markov Chains” and other works. The theoretical framing and entropy analysis largely restate standard Markov-chain properties without introducing new modeling insights or learning mechanisms.

2. Conclusions are largely descriptive/common-sense.
The observed divergence under sampling and convergence under greedy decoding are expected outcomes of temperature-based sampling. The paper’s central message—that iterative generation can yield stable or diverse outputs—is already well known and does not advance understanding beyond descriptive confirmation.

3. No validation on downstream tasks.
The study remains entirely at the sentence-level simulation stage. There is no attempt to connect the theoretical model to real-world LLM performance (e.g., degradation of reasoning, factuality, or coherence under iterative prompting). Thus, practical relevance is limited.

4. Missing model-based improvement.
The paper does not propose enough modification or algorithm that leverages the Markov-chain view to improve existing models or inference processes. Without an actionable outcome, the contribution is also limited.

**Questions:**

1. Could your framework predict or mitigate real LLM degradation phenomena (e.g., semantic drift, factuality loss) in iterative settings?

2. Are there quantitative validations on downstream benchmarks (translation quality, summarization consistency) demonstrating that the Markov formulation offers predictive or corrective value?

3. Can you provide any model modification or inference adjustment informed by your theoretical analysis?

---

> ### Author Response · Authors · 2025-11-21
>
> Thank you for your feedback and questions. We hope our answers could resolve your concerns.
>
> **Weakness 1**: Lack of novelty.
> **Answer**: We cited (in Line 333\) the paper you mentioned, Zekri et al. (2024) “Large Language Models as Markov Chains”, but **it is entirely different from our work**. Their study focuses on tokens, for example, using Markov chains to analyze repetitive patterns in the output of a single inference process. In comparison, our research centers on sentences, treating the output of one inference as the key element and examining the phenomena of multiple inferences (where each output becomes the input for the next inference). This is also why we used the term “Markovian Generation Chain” in the title. Therefore, our approach and interpretation are novel. We completely understand that you may not have enough time to read their paper, but we hope you won’t be misled by the two titles.
>
> **Weakness 2**: Conclusions are largely descriptive/common-sense.
> **Answer**: Similar to our response to the previous question, we also use Markov chains to explain this simulated phenomenon, which is not simply descriptive. The structure of LLMs is quite complex, and even simple ideas require simulation to be validated. For example, the cases in Tables 2 to 4 may be different from what some people might expect.
>
> **Weakness 3**: No validation on downstream tasks.
> **Answer**: We focus mainly on paraphrasing and translation in our research, as we consider these to be significant real-world applications for LLMs. Honestly, LLMs have a wide range of applications, and we cannot fully assess the impact of each one. In the past few days, we have done **new simulations on 450 paragraphs (ranging from 4 to 17 sentences)** across three datasets. For example, in simulations using GPT-4o-mini on paragraphs (temperature=0.7, top-p=0.9), certain sentences may repeat across different iterations. But it's uncommon for the whole paragraph to be repeated (i.e., all the sentences in the paragraph are the same). In the 50 iterations, the ratio of the number of sentence versions to the original number of sentences is 26.7 (BookSum), 19.7 (ScriptBase-alpha), 24.2 (News2024).
>
> **Weakness 4**: Missing model-based improvement.
> **Answer**: Thank you for your question. We believe our research provides researchers with a new perspective to evaluate the stability and creativity of models. We will include a discussion of it in the future version.
>
> **Question 1**: Could your framework predict or mitigate real LLM degradation phenomena (e.g., semantic drift, factuality loss) in iterative settings?
> **Answer**: Our framework and simulation can provide new insight, for example, degradation phenomena may only occur in the early stages of iteration, after which the system might stabilize or enter a cycle. This also means that after multiple iterations, the content could reach **a stable state rather than collapse**.
>
> **Question 2**: Are there quantitative validations on downstream benchmarks (translation quality, summarization consistency) demonstrating that the Markov formulation offers predictive or corrective value?
> **Answer**: We’re not completely sure we fully understand your question, but perhaps you can refer to **Figure 3 and Figures 8-12**, where we also measure the similarity of sentences before and after iteration. If there are fewer states (sentence versions) in the Markov chain, the loss of information is likely to be less.
>
> **Question 3**: Can you provide any model modification or inference adjustment informed by your theoretical analysis?
> **Answer**: Thank you for your question, but this may be beyond the scope of our paper. Our research offers a new perspective on the **features of LLMs**, but it does not suggest that this is a drawback of LLMs. In addition, we think that researchers can assess the stability and creativity of LLM outputs through sentence-level diversity, as it is at least an **intuitive and transparent metric**. We will include some related discussions in the next version.
>
> We welcome any further questions or comments you may have\!

---

> > ### Comment · Reviewer_svf8 · 2025-11-28
> >
> > Thank you for the thoughtful and detailed response, as well as the additional paragraph-level simulations. I appreciate the clarifications and the effort to expand the experiments. That said, my main concerns regarding the significance of the contribution and novelty, the limited experimental scope, and the connection to realistic downstream scenarios remain largely unresolved. The added results are helpful but do not substantially change my overall assessment of the work. I will therefore keep my original rating.

---

> ### Author Response · Authors · 2025-11-28
>
> Thank you for your response. However, we would like to emphasize once again that our settings have not been considered by others before. We would be very grateful if, despite your busy schedule, you could take a moment to look at Zekri et al. (2024) "Large Language Models as Markov Chains". Their story is completely different from ours. It just happens to have a similar name. You have the right to maintain the original score, but we hope that the misunderstandings in the review won't affect future readers, such as those who may refer to the reviews via openreview.

---

### Official Review · Reviewer_haCQ · 2025-10-31

**Soundness:** 4
**Presentation:** 4
**Contribution:** 4
**Rating:** 6
**Confidence:** 2

**Summary:**

The paper investigates what happens during iterative inference when a LLM's output is repeatedly fed back as its next input along with a fixed prompt. The authors term this process a "Markovian generation chain". They draw a clear distinction between this concept and "model collapse," which is a phenomenon of iterative training.

Through simulations on rephrasing and translation tasks, the study finds two main behaviors: (1) Greedy Decoding quickly converges to short, repetitive loops of sentences. (2) Sampling-based Decoding can generate a large number of distinct outputs, potentially increasing sentence-level diversity rather than reducing it.

The authors propose a theoretical framework based on Markov chains, where sentences are discrete states. The LLM's operation acts as the transition matrix. In this model, the loops seen in greedy decoding are communicating classes, while sampling-based decoding explores this state space stochastically, eventually converging to a stationary distribution.

**Strengths:**

1. **Novel and Relevant Problem:** The paper addresses a highly practical and under-studied question. While model collapse (iterative training) is well-researched, this focus on iterative inference chains mimics real-world scenarios where users repeatedly edit, translate, or rephrase content using LLMs.

2. **Strong Empirical Evidence:** The simulations are thorough, covering multiple models, different domains, and the two distinct tasks. The inclusion of Google Translate is also a particularly effective comparison.

**Weaknesses:**

1. **"Diversity" vs. "Factual Drift"**: The paper primarily measures diversity as the "number of unique rephrasings". However, the provided example in Table 2 (Appendix) shows the Llama-3.1 model's output not just rephrasing "We begin with a prologue" but progressively adding new, unprompted information. The paper notes this as "information in the sentences having changed", but doesn't critically analyze this as a potential failure of semantic preservation (which the prompt explicitly requested ). It's unclear how much of the measured "diversity" is faithful paraphrasing versus hallucination and semantic drift.

2. **Applicability to real-world scenerios:** The simulations make sense, but it is hard to map this to any real world use case where a user will not repeatedly ask the same prompt. A genuine human-in-the-loop process  would involve user edits or, more likely, changing prompts at each step. This would create a non-stationary process, which the paper's static Markov model does not account for.

**Questions:**

1. **Explaining Model-Specific Behavior:** You show models behave differently in Figure 2, where Llama and Mistral are far more "creative" than GPT-4o-mini and Qwen. Qwen even appears to enter a loop under sampling. Why do you believe this is? Is it a result of different sampling parameters, pre-training data, or perhaps the fine-tuning process (e.g., some models being RL'ed into a lower-entropy, more stable state?

---

> ### Author Response · Authors · 2025-11-21
>
> Thank you for your thoughtful summary and insightful questions.
>
> **Weakness 1**: "Diversity" vs. "Factual Drift"
> **Answer**: Thank you for highlighting this point. We also agree with your view that LLMs can produce factual drift during text processing, even though we’ve included the instruction in the prompts to “rephrase it while preserving all the original meaning and without losing any context”. Table 2 in the appendix is meant to help illustrate this point. But our focus is on exploring the real-world impact, where this type of drift often arises during LLM usage. **Strictly maintaining consistency between input and output might not truly reflect the real-world dynamics.** We will also provide more critical analysis in the next version based on your questions.
>
> **Weakness 2**: Applicability to real-world scenarios
> **Answer**: Thank you for raising this point. We also think that the same user generally wouldn’t use LLMs with identical prompts to process the same text multiple times. We have already mentioned it in the paper (L438-L441). What we’re trying to model is the situation where different users apply the same prompts to the same text repeatedly, such as in translation tasks, which is something that often happens in real-world applications. We have also supplemented the simulation with **alternating rephrasings using different prompts**, and the results are similar, where cycles appear but the states (sentences) differ within those cycles. We will update these results in the next version. In the past few days, we have done **new simulations on 450 paragraphs (ranging from 4 to 17 sentences)** across three datasets. For example, in simulations using GPT-4o-mini on paragraphs (temperature=0.7, top-p=0.9), certain sentences may repeat across different iterations. But it's uncommon for the whole paragraph to be repeated (i.e., all the sentences in the paragraph are the same). In the 50 iterations, the ratio of the number of sentence versions to the original number of sentences is 26.7 (BookSum), 19.7 (ScriptBase-alpha), 24.2 (News2024).
>
> **Question**: Explaining Model-Specific Behavior
> **Answer**: Thank you for your question. Given the complexity of modern LLMs, it is difficult to provide a fundamental explanation, and we can only describe the results of simulations and make speculative assumptions. We think fine-tuning is a more likely possibility.
>
> We welcome any further questions or comments you may have\!

---

> > ### Comment · Reviewer_haCQ · 2025-11-26
> >
> > Thank you for the detailed reply. I will keep my score.

---

### Official Review · Reviewer_3Zk2 · 2025-10-31

**Soundness:** 1
**Presentation:** 2
**Contribution:** 1
**Rating:** 2
**Confidence:** 4

**Summary:**

Given a passage, the model is prompted to produce a semantically equivalent paraphrase. The output is then recursively re-input for a total of 50 iterations. The paper analyzes what happens in this sequence, such as how many unique paraphrases are generated and how diverse the generated paraphrases are. While I appreciate the authors’ efforts at studying this phenomenon, I fail to understand why this process is significant at all and worth studying. The question of figuring out what happens when models are iteratively trained on its generated data recursively, which is studied in existing work and the authors point it out too, is clearly an important question because people use LLMs to generate text which is then put on the internet and becomes the training data for next iteration of frontier LLMs. But for the case of paraphrasing (inference) which is studied in this work, I don’t see why it is important and I don’t see a clear augment in the paper for why it is important either. I was also unable to see the precise contribution of the work from its introduction or abstract, and was unsure about what to take away from statements like “Researchers need to think more about where this chain will lead.”, “We also seek to explore the wider and deeper impact of LLMs on language and society.”

**Strengths:**

The paper carries out extensive analysis of how the paraphrases (when generated repeatedly by prompting the model on previous outputs) compare with each other, such as number of unique paraphrases and their textual similarity measured via automated metrics like BLEU and ROUGE, while using multiple LLMs. However the purpose of this analysis remains unclear to me.

**Weaknesses:**

As mentioned in the summary section

**Questions:**

NA

---

> ### Author Response · Authors · 2025-11-21
>
> Thank you for your concise feedback and questions. We hope the following answers will address your concerns, and we will also provide more clarifications and additional information in future versions.
>
> 1. This process can be considered a case of multi-turn interactions with LLMs, which should be a very important research topic. For example, there is a workshop on this very theme at NeurIPS this year: NeurIPS 2025 Workshop on Multi-Turn Interactions in Large Language Models ([https://workshop-multi-turn-interaction.github.io/](https://workshop-multi-turn-interaction.github.io/)).
> 2. The simulations and analysis we conducted are **not model collapse**, although there are some connections. As model collapse occurs when data generated by LLMs is reused for training, we examine what happens when LLM-processed data is reintroduced to LLMs for further processing.
> 3. In addition to analyzing paraphrasing, we also examined the case of translation. We believe that the **two scenarios we focus on are more common (and maybe also more important) in the real world than model collapse**, but research on them is currently lacking. In reality, the same content can be easily iterated multiple times during LLM inference by different users. This can happen anytime, anywhere, with almost no barriers. But for model collapse, iterations involving new and old models obviously take much longer in the real world, and only a few professional players capable of training LLMs are at the table.
> 4. In the past few days, we have done **new simulations on 450 paragraphs (ranging from 4 to 17 sentences)** across three datasets. For example, in simulations using GPT-4o-mini on paragraphs (temperature=0.7, top-p=0.9), certain sentences may repeat across different iterations. But it's uncommon for the whole paragraph to be repeated (i.e., all the sentences in the paragraph are the same). In the 50 iterations, the ratio of the number of sentence versions to the original number of sentences is 26.7 (BookSum), 19.7 (ScriptBase-alpha), 24.2 (News2024).
> 5. The term “chain” in the sentence “Researchers need to think more about where this chain will lead” is used as a double meaning: it refers both to the Markovian generative chain mentioned in the title, and to the chain reaction that results from the real-world application of LLMs. “We also seek to explore the wider and deeper impact of LLMs on language and society” refers to our long-term goal, and we will refine the expression of this statement.
>
>
> We welcome any further questions or comments you may have\!

---

### Author Response · Authors · 2025-12-04

Dear reviewers, ACs, and SACs,

We thank the reviewers for their valuable feedback. Unfortunately, due to well-known reasons, our communication with the reviewers is insufficient. We have updated the paper and would like to emphasize the following points:

* We have added simulations using alternating prompts and also included simulations for the paragraphs rather than just single sentences. Our conclusions remain unchanged.
* While it may seem straightforward in terms of sentence diversity, very few studies have explored this topic, and none has analyzed the impact of LLM-processed sentences on human language corpora through the lens of Markov chains.
* Based on the reviewers’ suggestions and concerns, we have modified some expressions and added relevant clarifications.

Best,

---

### Meta-Review · Area_Chair_ZEWv · 2025-12-16

**Summary:**

The paper studies what happens when you repeatedly "feed" the output of an LLM back to itself as a prompt, and attempts to characterize what the convergence of this process is in the limit --- as a function of both the chosen LLM, and various inference hyperparameters like temperature / greediness.

I agree with the consensus of the reviewers that the paper needs more work to meet the ICLR bar. The motivation for this question is pretty slim / weak (as opposed to related, but distinct questions like model collapse); the conclusions are rather expected (greedy collapses faster, more entropy in decoding collapses slower); and what  the Markov Chain "analysis" is supposed to show is very unclear (the section largely just restates some standard properties of discrete Markov Chains).

**Reviewer Concerns:**

I do not think the main concerns stated in my meta review (weak motivation; straightforward/expected conclusions; unclear value/goal of the Markov Chain lens) were addressed. All these weaknesses were mentioned in some form by all the reviewers.

**Reviewer Scores:**

I do not think the scores would have moved --- I do not think the answers by the authors adequately address the concerns mentioned by the reviewers and reiterated in my meta-review.

---

### Decision · Program_Chairs · 2026-01-26

Reject